# Comparing Children’s Behavior Problems in Biological Married, Biological Cohabitating, and Stepmother Families in the UK

**DOI:** 10.3390/ijerph192416543

**Published:** 2022-12-09

**Authors:** M. Rachél Hveem, Samuel C. M. Faulconer, Mikaela J. Dufur

**Affiliations:** Department of Sociology, Brigham Young University, Provo, UT 84602, USA

**Keywords:** family structure, stepmothers, cohabitating, internalizing behavior problems, externalizing behavior problems

## Abstract

A large body of research shows that children who live with two married biological parents have lower levels of externalizing and internalizing behavior problems compared to their peers in other family structure, including cohabitating biological families. Such patterns suggest that marriage provides a uniquely protective family environment, though we know less about children in the obvious counterfactual case: married stepfamilies. While research suggests children with stepfathers have more behavior problems than those living with married biological parents, we know little about how children with stepmothers fare, or how children with stepparents fare compared to those living with cohabiting biological parents. We use the Millennium Cohort Study (MCS) sweep 6 to compare children living with married biological parents, married fathers and stepmothers, and cohabiting biological parents. We find that family structure has no significant relationship with children’s internalizing behavior problems, but that children living with a stepmother and biological cohabitating families exhibit more externalizing behavior problems than do those living with married biological parents. Covariates that indicate both physical and social family environments must be considered together to explain differences in married-parent families on externalizing behavior problems.

## 1. Introduction

Since the publication of their seminal work *Growing Up With a Single Parent: What Hurts, What Helps*, McLanahan and Sandefur’s foundational work showing consistent associations between living with two married parents and positive child outcomes has been cited more than 5500 times, establishing a strong body of research demonstrating an association between certain family structures and positive child outcomes [1] (see also [2]) The effects of these family environments on children’s behavior start as early as age 2 [3]. A large body of research demonstrates positive and protective effects of living with two biological parents who are married to each other, and these positive effects span to a wide variety of children’s outcomes [4,5,6,7]. For example, children in biological married families tend to have fewer behavior problems compared to all other family structures [8,9], whereas children in single-parent families have more behavior problems compared to other family structures [5,10,11,12]. Work comparing children who live with married biological parents to other family structures suggests unique effects both of marriage and of living with biological mothers [13,14], but more research is needed to disentangle these components of the family environment and their effect on child health and development.

Our work contributes to the existing literature in two important ways that consider the nuances of parental relationships to each other and to their children. First, we examine how parental legal relationships to each other might be related to child outcomes by comparing children of married and cohabitating parents in a context in which cohabitation is common and, therefore, living in a cohabitating family might carry less stigma than in other settings. Second, we examine how parental biological relationship to child might be related to child outcomes by using a large, nationally representative dataset that allows a sufficiently large sample of children living with stepmothers that we can compare to children living with married biological mothers and married nonbiological mothers.

### 1.1. Children’s Behavior Problems

A large and continually growing literature demonstrates that family structure has a significant association with children’s outcomes [1,2,12,15], but more recent research has moved toward attempting to explain whether these associations are the result of the family structure itself or of resources or mechanisms related to but independent from the family structure. We add to this literature by examining nuances of legal and biological relationships in the context of an outcome that is strongly related to both short- and long-term child attainment and health: child behavior problems. Increased behavior problems are associated with drug use at a young age [16]. Behavior issues lead to lower educational attainment and employment outcomes later in life [17,18]. Increased internal and external behavior issues are found to lead to early mortality [15]. Family structure has a significant effect on children’s behavior problems [19]; as a result of this, it is important to understand the impact of behavior problems among adolescents to address behavior problems and prevent negative outcomes in the future. 

### 1.2. Biological Familiy Enviornment

The family environment and changes to that environment have a direct impact on children’s behavior problems. Children in married biological families have fewer externalizing behavior problems than children in other family structures, as well as better overall health. Children in all biological families have fewer mental health problems [20]. The differences in children’s internalizing behavior problems seems less tied to family structure, but is sensitive to changes in family structure that increases family stress [3,4,8]. 

In addition to family structure, family environment may include other factors such as parental warmth. Parental warmth is created through positive parent–child interaction. Children who live in warm family environments tend to have better child social adjustment scores, and thus are less likely to have externalizing behavior problems than their peers [2,3,19]. Parental warmth and quality of relationships vary across family structure. Family structures outside of two-parent biological families have lower rates of parental warmth [20]. When comparing cohabitating biological families to married biological families, parental warmth is highest in married biological families [14]. Higher rate of parental warmth in married biological families continues to explain why children in married biological families have lower rates of behavior problems. 

However, more nuanced research pushes us to ask whether these observed relationships between living with married parents and child outcomes are due exclusively to the structure of the family or whether these patterns can be better explained by mechanisms related to family structures. For example, research connecting parent sex with child behavior by looking at two-parent and single-parent families may be conflating parent sex with the number of parents available to children [21,22]. Similarly, some research suggests that married parents on average enjoy advantages in financial and social resources that explain better outcomes among their children [23,24]. 

### 1.3. Biological Cohabiting Family Environment

We extend such research by comparing the effects of marriage and biological relationships with children. For example, children who live with both biological parents in a cohabiting family should enjoy many of the same benefits that children in married biological families have access to. For example, evolutionary biology theories would predict that biological parents would invest heavily in their offspring’s health and development regardless of whether they are married to the child’s other parent [25]. If biology is the primary factor influencing parental investment in children and family environments, children living with their biological parents who cohabit should have similar outcomes to their peers who live with biological parents who are married. 

However, parental cohabitation is associated with, at best, mixed outcomes for their children. Children in cohabiting families are significantly less engaged in school and have lower levels of achievement compared to married families [14,26]. Looking specifically at behavior, similar models across similar large data sets provide confounding results. Some research claims that children from biological cohabitating families seem to have similar behavior scores to children in married biological families [27,28,29]. However, as was the case for academic outcomes, a large body of research finds the opposite, as children who are raised in cohabitating families do worse than those in married biological families in relation to behavior problems [14,27,28,30,31]. It remains unclear the degree to which parental marriage affects children’s behavior problems when children are living with both biological parents. 

### 1.4. Stepparent Family Enviornment 

There is also a question of whether the gender of biological parent available to the child is important. Much research comparing children living with one biological parent and one stepparent has found that children who lack access to one of their biological parents do worse across a number of outcomes than do children who live with both biological parents [32,33,34,35,36]. However, these studies tend to compare children living with two married biological parents to those living with a biological mother and stepfather, potentially conflating important effects of marriage or parental gender.

The increased behavior problems that are present in stepfamilies are often explained by the increased stress that children experience from family dissolution and/or reconstitution [37]. In families where stepfamilies have been created as a result of an initial divorce that separates the biological parents, the effects of this stress are seen when comparing children’s scores before and after divorce. After divorce children have lower academic achievement, more behavior problems at school, and lower rates of self-esteem [31,38]. However, family environment changes are not the only contributor to differences in stepparent families; stepparent–child relationships also influence children’s behavior problems. Positive stepchild-stepparent relationships are associated with lower externalizing behavior problems and higher scores of self-esteem among females [39], but children report lower-quality relationships on average with the stepparent than with the biological parent [37]. Improving stepparent-stepchild relationships may protect against the impact family disruption has on children’s behavior outcomes. 

### 1.5. Stepmother Family Enviornment 

The ability to improve stepparent-stepchild relationships, however, may be dependent on gendered roles and expectations in families [40,41]. As a result, it is important to look specifically at stepmother families. Stepmother families are fewer in number than stepfather families, and certainly fewer in number than married biological families, often due to gendered societal and legal expectations of biological mothers being superior parents [42]. Because of this, fathers receiving custody and being able to create stepmother families may indicate particularly stressful family dissolutions [11,43]. At the same time, legal shifts in recognizing fathers’ rights and joint custody options are resulting in increases in stepmother families. Despite this increase, there is limited research on the effect stepmothers have on children’s behavior. 

Stepmothers sometimes have a lack of understanding of their role within the family, which can limit their interaction with their spouse’s children and influence the children’s outcomes. Friction in trying to define their role in the family can also be influenced by lack of clarity in the expectations set by the biological father, which can be harmful to the stepmother’s mental health and in turn affect the success of the family [44,45]. Stepmothers sometimes feel the need to act “like a mother,” with those actions based on typical gender roles and stereotypes, but feel unable or reluctant to fill this role of a “typical mother” because of the presence of a non-residential biological mother in children’s lives or because children are unsure how to treat a nonbiological mother [46]. This can increase mental health challenges for stepmothers, possibly resulting in more family stress compared to other family environments [12,37,47]. This in turn might result in higher behavior problems among children.

As a result of this lack of clear roles and expectations, much of the current research on stepmothers’ effect on children has mixed outcomes. Some research has found that stepmothers have essentially no impact on their stepchildren, but rather that children’s behavior problems likely stem from the higher stress of initially entering single-father families and undergoing additional family transitions [39,47]. In cases where moving into a single-father family is related to particularly traumatic circumstances, having a stepmother present can even have a positive influence on children’s behavior [48]. The majority of the scant research on stepmothers has found a negative relationship between living with a stepmother and children’s behavior. Poor relationships between children and stepmothers result in lower inhibition and higher aggression ratings, and female children in these circumstances report lower self-esteem [39]. The impact of having a stepmother seems to affect daughters more strongly than sons, with daughters in stepmother families often participating in more risky sexual behavior [7]. 

It is also possible that resources explain differences in child behavior problems across family structures. While children in all three of the family structures we study here have access to two adults in the home, those living with cohabiting parents tend to have lower incomes than those in two-parent married families, which is in turn related to worse cognitive and social-behavior development [27,49]. While stepmother families may have more economic resources than cohabiting families, research on stepfathers suggests that stepparents reserve some resources for nonresident family members. On the other hand, we would expect children living with two biological parents, regardless of their marital status, to have access to similar amounts of family social capital. However, youth in stepmother families may have access to less social capital since ties between stepmothers and stepchildren may be weaker due to unclear roles and expectations. Given the strong connections between family social capital and child behavior [50], understanding such differences could help to explain the behavior differences across family environments.

### 1.6. Current Study

Research on specific family environments allows for a greater understanding of how family environments affect child behavior problems. Research in the past has analyzed primarily stable (married biological) vs. unstable families (all other family types) [4,5,51]. We extend those inquiries here to examine both the effects of marriage by including biological cohabiting parents, but also the effects of parent sex, by examining children in stepmother families. Stepmother families, in particular, have been understudied as a result of the limited data including stepmothers [14,27]. 

In this study, we extend previous research adjudicating between explanations focusing on family structure and mechanisms associated with family structure by analyzing children’s internalizing and externalizing behavior problems across three family types: living with two married biological parents, living with two cohabiting parents who are not married, and living with a biological father and a stepmother. Our work contributes to the existing literature in two important ways. First, by using data on children from the United Kingdom in the Millennium Cohort Study, we examine a context in which cohabitation is more common than in many other previously studied settings and, therefore, living in a cohabitating family might carry less stigma, to see if previously observed negative relationships between living with cohabiting biological parents and child outcomes persist in a high-cohabitation setting. Second, by using a very large, nationally representative dataset we are able to derive a sufficiently large sample of children living with stepmothers to compare children living with married biological mothers, cohabitating biological mothers, and married nonbiological mothers. Taken together, these allow us to further examine whether nuances in parents’ legal relationships with each other or biological relationships with children might help explain patterns of differences in child outcomes across family structures.

Data from the UK provides a unique perspective on cohabiting families. In the United States, where much of the family structure and family environment literature has been derived because of access to large-scale datasets, only 3% of women and 2% of men cohabitate. In the UK, the rate of cohabiting is substantially higher. 10% of women and 12% of men cohabitate with their partner [52]. These rates are similar for parents. The higher rates of cohabitation in the UK allow us to gain a better understanding of the cohabiting family environment and the impact it has on children’s behavior problems. In addition, the Millennium Cohort Study (MCS) data we use here comprise a large enough sample that we are able to perform multivariate models examining stepmother families at the same time. We speculate there may be a “hierarchy” of family environment effects, with children in stepmother families reporting the most behavior problems, those in married biological families reporting the fewest, and children from cohabiting biological families in between, because they live with their biological mothers but do not enjoy the potential benefits of legal protections or social acceptance of living with married parents.

## 2. Material and Methods

### 2.1. Sample

The participants for this study come from the Millennium Cohort Study (MCS), a longitudinal study that looked at more than 21,000 children born across the United Kingdom (England, Wales, Scotland, and Northern Ireland) in 2000. The Millennium Cohort Study was collected by The Centre for Longitudinal Studies at the University College London under the auspices of the National Health Service Research Ethics Committees in South West, London, and Yorkshire, from whom they received ethical approval. We obtained access to the MCS data through contract with the UK Data Services. This analysis focuses on Sweep 6, which looks at 18,818 children who are 14 years old at the time the data was collected. Parents report on their family context, parental environment, parenting, child and parental health, and children’s behavior and cognitive development; children also fill out a survey. Sweep 6 is the first sweep within this study to have enough stepmother families to include in multivariate analyses. We study the UK in part because there is a higher proportion of cohabiters than in countries like the US from which much of the family structure literature is derived [18,52,53]. This study excludes those living with parents who are single mothers, single fathers, grandparents, parents in civil unions, stepfathers, and cohabitating stepfathers. After these exclusions, the analytic sample is 7182, with 6521 two-parent married families, 589 two-parent cohabiting families, and 77 two-parent stepmother families. Given our focus on marriage, it is unfortunate that we have too few married and cohabiting stepmothers to split into the two marriage contexts to analyze; however, it is useful to know that a large majority of the stepmothers included here are married (n = 58) and 19 of the stepmother families are cohabitating. Because the MCS sample was initially collected at the time of the target child’s birth, almost all of the children living with two parents are living with heterosexual couples. In our initial sample creation, we identified one couple where two women reported living in a civil union; we excluded this family from the sample because there were too few families in this group to draw conclusions. 

### 2.2. Measures

We use Sweep 6’s parent-reported data to measure child behavior problems. In Sweep 6, parents answer several questions regarding their child’s behavior and emotional problems as part of the Strengths and Difficulties Questionnaire (SDQ) [54]. Following documented methods from the MCS [55], we used this questionnaire to create two scales that measure internalizing and externalizing behavior problems. The internalizing behavior problem scale is made up of 10 variables, while the externalizing behavior problem scale is made up of 12 variables. These two scales are our outcomes for this study.

Internalizing behavior problems are measured through emotional symptoms and peer problems. Emotional symptoms were composed of the following variables: target child often complains of sickness, often seems worried, often feels unhappy or tearful, is nervous or clingy in new situations, and is easily scared. Peer problems are composed of the following variables: target child prefers to be alone, has at least one good friend, is liked by other children, is bullied by others, and gets along better with adults than children (*a* = 0.77).

Externalizing behavior problems are measured via conduct problems and hyperactivity. Conduct is based on the following variables: does target child often have temper tantrums or have a hot temper, target child is obedient, target child usually does what adults request, does the target child often fight with other children or bully them, does the target child steal from their home, school, or elsewhere, does the target child often lie or cheat. Hyperactivity is based on children’s traits of restlessness, fidgeting, being easily distracted, thinking through decisions (reverse coded), and seeing tasks through to completion (reverse coded) (*a* = 0.76). 

Our primary independent variable is family structure. We constructed the family structure types using the main respondents’ responses regarding their marital status and relationship to the cohort member/target child as reported in Sweep 6. We also use the MCS-derived variable that analyzes the relationship between cohort members and the parents/caregivers in the household. We created three family structures: married two-biological-parent families, cohabiting two-biological-parent families, and two-parent stepmother families. 

We also included several control variables that help to make up family environments and have shown associations with child behavior problems in previous research. We enter these in theoretical blocks. The first refers the social resources children have available to them in their families. This includes a measure of how close the target child is to their mother and their father, each on a 0–4 scale where higher numbers indicate more closeness, and the primary parent’s reports of overall closeness to the target child on the same scale. For the child questionnaire, the questions reads “overall how close would you say you are to your mother.” For the parent questionnaire, the question reads, “overall how close would you say you are to [cohort member’s name]?” For both questions responses range from “not very close” to “extremely close,” 

The next theoretical block taps economic resources. These include MCS’s measures of whether or not the family owns a home (0 = no, 1 = yes), family income measured in quintiles [14,45,50], and whether or not the family receives child benefits (0 = no, 1 = yes).

The next block includes parental health and behavior variables. These includes individual items from the MCS measuring whether the parent smokes or drinks (0 = no, 1 = yes), parental general health (analyzed through the following options: 1 “Excellent” 2 “Very good” 3 “Good” 4 “Fair” 5 “Poor” health), and parental depression, which is a dichotomous variable asking if the parent reports having severe depression. 

The next theoretical block includes child characteristics. Children who have more siblings are more likely to have internalizing behavior issues [9,56]. We truncated the original MCS variable on the number of siblings the target child has at five to reflect the distribution in the data (0–5 siblings). We also include a measure of the child’s general health. This variable is a five-item scale created by the MCS where 1 “Excellent” 2 “Very good” 3 “Good” 4 “Fair” 5 “Poor” health. We include a dichotomous measure of child’s involvement in extracurricular activities (0 = no, 1 = yes) and a an original MCS variable (ranging from 0–4) tapping how often the child uses marijuana, tobacco, alcohol, or e-cigarettes, with higher values indicating more drug use. These activities help to understand children’s social involvement [57]. 

Finally, we speculate that cohabiting and stepmother families might experience more family stress because of their family structures, and family stress has been associated with strong negative impacts on both family environments and child outcomes [4,18,19,34,56]. High family stress is associated with more behavior problems. We created a family stress scale made up of five variables: experiencing mental illness in the family, experiencing eviction, exposure to alcohol use in the family, experiencing financial difficulties, and unemployment. Measurement and descriptive details for all variables can be found on Table 1.

### 2.3. Missing Data

Respondents who were missing the data required to create our family structure variable were dropped. Other missing data was replaced using Stata 16’s MICE multiple imputation protocol. We imputed data for cohort member drug use, cohort member health, extracurriculars, parent reported closeness with child, parent depression, parent drinking, and internalizing/externalizing behavior problems. We use the 20 imputed data sets in the analyses below.

### 2.4. Analytic Plan

We first examine mean or proportion of each variable in the model for our three focal family structures to get a descriptive sense of potential differences across family environments. We then use a series of seven stepped OLS regression models to test our theoretical blocks. Model 1 analyzes only the relationship between family structure and behavior problems, excluding all controls. In Model 2, we add controls for parental closeness. Model 3 removes parental closeness and includes economic resources. Model 4 removes economic resources and includes parent health and behavior variables. Model 5 removes parental health variables and includes child characteristics and behaviors. Model 6 removes child characteristics and adds the family stress scale. The final model includes all theoretical blocks and all controls for a full model. Each model, then, tests the relationship between family structures and child behavior problems, while accounting for different potential theoretical explanations.

We repeat these analyses four times. First, we predict internalizing behavior problems with stepmother families as the reference group (compared to cohabiting two-biological-parent families and married two-biological parent families). We then predict externalizing behavior problems using stepmother families as the reference category. We then repeat these analyses using cohabiting families as the reference category (compared to stepmother families and married two-biological parent families) for internalizing and externalizing behavior problems.

## 3. Results

### 3.1. Descriptive Analyses

Table 2 contains descriptive statistics broken up by family structure type. This table helps illustrate how stepmother, biological married, and biological cohabiting families may differ in important ways. Looking at these averages, stepmother families and biological cohabitating families are more similar to each other than to married biological families. For example, 85% of biological married families own a home compared to only 60% of biological cohabiting families and 66% of stepmother families. When looking at family stress, biological married families have the lowest levels of stress at only 0.386, compared to stepmother families with the highest level of stress (0.429) and biological cohabiting sitting between those averages (0.403). Surprisingly, biological married and biological cohabiting families were closer to each other when looking at family income. Child health and parental health has similar averages across all three family environments, though stepmother families, and especially biological cohabiting families, report more parental depression than do biological married families. Youth in stepmother families report substantially lower levels of closeness to mothers, though parents in each family structures reported similar levels of closeness to their children.

### 3.2. Internlaizing Behavior Problems and the Case of Stepmothers

In Table 3, we examine internalizing behavior problems. Comparing family structures to stepmothers, neither children from biological cohabiting nor biological married families are statistically different on internalizing behavior problems. This provides no evidence for our predicted “hierarchy” with children in stepmother families having the most behavior problems, though we note the small number of stepmother families may affect our ability to find significant relationships. These family structure patterns, with children in stepmother families not having statistically different on internalizing behavior problems than those in cohabiting and married biological families, persists when all of our potential theoretical explanations are included.

Aside from not affecting the relationships between family structure and internalizing behavior problems, controls behave largely as expected. Model 2 shows that each of our measures of family closeness has a statistically significant and negative association with internalizing behavior problems. Children who are close to their father (b = −0.488, *p* < 0.001) or their mother (b = −0.201, *p* < 0.001) tend to have fewer internalizing behavior problems, as do children whose parents report being closer to them (b = −0.306, *p* < 0.001). Model 3 suggests that home ownership (b = −0.337 (*p* < 0.001) and income (b = −0.434, *p* < 0.001) both have a significant effect on behavior. Owning a home and having a higher income are associated with fewer internalizing behavior problems. Receiving child benefits was not significantly associated with internalizing behavior problems.

Models 4 and 5 look and health variables for the parent and the child. Looking at parental health, parental smoking (b = 0.428, *p* < 0.001), parental drinking (b = 0.453, *p* < 0.001) poorer overall health (b = 0.282, *p* < 0.001), and depression (b = 0.212, *p* < 0.001) are all positively correlated with internalizing behavior problems among children. Children’s health and behaviors also show similar significant results: more siblings (b = 0.169, *p* < 0.001) and children’s overall poorer health (0.638, *p* < 0.001) are positively correlated with internalizing behavior problems. Involvement in extracurricular (b = −1.942, *p* < 0.001), and drug use (b = −0.166, *p* < 0.001) are both negatively correlated with internalizing behavior problems. Controls behave as anticipated. Model 6 includes the family stress scale, finding that higher rates of stress have a statistically significant and positive association with children’s internalizing behaviors (b = 0.739, *p* < 0.001). 

Model 7, the full model, analyzes all control variables and their impact on internalizing behavior health problems. The results suggest that parental smoking (b = 0.225, *p* < 0.001), parental health (b = 0.150, *p* < 0.001), parental depression (b = 0.191, *p* < 0.001), child’s health (b = 0.433, *p* < 0.001), having more siblings (b = −0.164, *p* < 0.001), and family stress (b = 0.186, *p* < 0.001) continue to be positively associated with children internalizing behavior problems. Maternal closeness (b = 0.135, *p* < 0.05) has a positive association on children’s behavior problem in the presence of other controls, though the effects of father closeness and parent-reported closeness to child remain negative (father closeness b = −0.300, *p* < 0.001; parent-reported closeness b = −0.288, *p* < 0.001). Higher income (b = −0.255, *p* < 0.001), involvement in extracurriculars (b = −1.755, *p* < 0.001), and child’s drug use (b = −0.249, *p* < 0.001), result in fewer behavior problems. While the negative coefficient for child drug use seems odd on its face, we speculate that children who experience more mental health challenges might self-medicate with illegal drugs.

Throughout all 7 models, family environments do not show statistically significant differences. Differences in children’s internalizing behavior problems can be explained through father closeness, mother closeness, general parental closeness, income, parental smoking, parental health, parental depression, sibling number, children’s health, child drug use, as well as children’s involvement in extracurriculars.

### 3.3. Externalizing Behavior Problems and the Case of Stepmothers

Table 4 examines the relationship between family structure and child externalizing behavior problems. The results of Model 1 suggest that children’s externalizing behavior problems vary based on the family environment. Comparing family structures to stepmothers, children from married biological families (b = −1.369, *p* < 0.001) have fewer externalizing behavior problems. The difference between stepmother families and cohabiting biological families, however, is one-third the size, and is not statistically significant. This gives some support for our predicted “ordering” of behavior problems, but only for the idea that children in married biological families would do best. There is no evidence from these analyses that children in cohabiting biological families have fewer externalizing behavior problems than do children in stepmother families. This also provides some evidence of marriage having a protective effect for children, but only for marriage between biological parents. 

Model 2 adds measures of parental closeness, which are significant predictors of children’s externalizing behavior problems. High paternal closeness (b = −0.177, *p* < 0.001), maternal closeness (b = −0.234, *p* < 0.001), and parent-reported closeness (b = −0.859, *p* < 0.001) are all associated with fewer externalizing behavior problems. After controlling for parental closeness, children from biological married families (b = −0.921, *p* < 0.05) still show significantly fewer externalizing behavior problems compared to those in stepmother families. Model 3 analyzes economic resources. Children have fewer externalizing behavior problems when their parents own a home (b = −0.510, *p* < 0.001), as well when they report higher income (b = −0.452, *p* < 0.001). After controlling for economic resources, children from biological married families still have fewer externalizing behavior problems than children from stepmother families (b = −0.873, *p* < 0.05). The difference between stepmother families and cohabiting biological families remains nonsignificant.

Models 4 and 5 look at health variables for the parent and child. Looking at parental health, parental smoking (b = 0.860, *p* < 0.001), drinking (b = −226, *p* < 0.05), overall health (b = 0.216, *p* < 0.001), and depression (b = 0.177, *p* < 0.001) all have a statistically significant and positive associations with children’s externalizing behavior problems. The difference between married biological and stepmother families remains statistically significant. Children’s health and characteristics have a similar impact: having more siblings (b = 0.308, *p* < 0.001), children’s overall health (b = 0.255, *p* < 0.001), and drug use (b = 0.584, *p* < 0.001) are positively correlated with externalizing behavior problems, whereas involvement in extracurricular (b = −0.686, *p* < 0.01) is associate with fewer externalizing behavior problems. Married biological and stepmother families also remain significantly different on externalizing behavior problems after controlling for child health (b = −1.014, *p* < 0.01).

Model 6 includes the family stress scale, finding that higher rates of stress (b = 0.544, *p* < 0.001) are associated with higher levels of children’s externalizing behaviors. Children in stepmother families continue to have significantly more externalizing behavior problems compared to children in married biological families’ children (b = −1.346, *p* < 0.001); they continue, however, to not differ significantly from children in cohabiting biological families. 

Model 7, or the full model including all theoretical bocks, shows the impact family structure has on externalizing variables when all control variables are analyzed. When all variables are included in the model, there is no longer a statistically significant differences between children in stepmother families and those in married biological families. Supplemental analyses (available upon request) demonstrated that the stepmother effect was rendered nonsignificant by including both social resources and economic resources in the model at the same time. Poor parental health continues to be positively associated with externalizing behavior problems, while parental closeness continues to exert protective effects. Parental smoking (b = 0.417, *p* < 0.001), parental health (b = 0.122, *p* < 0.01), parental depression (b = 0.448, *p* < 0.001), and child’s drug use (b = 0.491, *p* < 0.001) all have a negative impact on children externalizing behavior problems. Children who are close to their parents (b = −0.782, *p* < 0.001), whose families own a home (b = −0.242, *p* < 0.05), and whose families have a higher income tend to have fewer external behavior problems. These results suggest initial support for the idea that stepmother families are at a deficit compared to married biological families in terms of externalizing behavior problems, but when stepmother families have similar social and economic resources as married biological families, their children do not fare worse.

### 3.4. Internlaizing Behavior Problems and the Case of Biological Cohabiters

Table 5 examines the relationship between family structure and child internalizing behavior problems, but with cohabiting biological families as the reference group to test differences between that family environment and married biological families. The results of Model 1 show again, as was true in Table 3, no statistically significant differences in internalizing behavior problems between cohabiting biological and stepmother families. However, children in cohabiting biological families have significantly more internalizing behavior problems than do those in married biological families (b = −0.621, *p* < 0.001). This provides evidence of marriage having a protective effect for children, but only for marriage between biological parents. 

Model 2 adds measures of family closeness. In the presence of mother closeness, father closeness, and parental reports of closeness, the statistically significant differences in internalizing behavior problems between cohabiting biological families and married biological families persist (b = −0.342, *p* < 0.05). The same is true when entering controls for family resources (Model 3), parent behaviors and health (Model 4), child health and characteristics (Model 5), and family stress (Model 6). However, in the full model (Model 7), the difference between cohabiting biological families and married biological families no longer reaches statistical significance. Supplementary analyses (available upon request) suggest that taking into account family closeness and economic resources in the same model explains away differences in child internalizing behavior problems between those two family structures. Internalizing behavior problems are similar for children in cohabiting biological families and stepmother families across all models.

### 3.5. Externlaizing Behavior Problems and the Case of Biological Cohabiters

Turning to child externalizing behavior problems (Table 6), there is again no statistically significant difference between cohabiting biological families and stepmother families, and this pattern persists across all models. By contrast, children from married biological families have significantly fewer externalizing behavior problems than those in cohabiting biological families in the bivariate (b = −0.960, *p* < 0.001) (Model 1). As was true when looking at internalizing behavior problems, this provides evidence of marriage having a protective effect for children, but only for marriage between biological parents. A statistically significant difference in favor of married biological families persists across all models, meaning that even after we control for family closeness, economic resource, parent health and behavior, child health and characteristics, and family stress, children in cohabiting biological families have significantly more externalizing behavior problems than do children in married biological families.

## 4. Discussion 

We posed questions concerning how exposure to marriage and to biological mothers would be associated with children’s behavior problems in the UK. While there is some evidence for the idea that lack of access to biological mothers may be associated with negative child outcomes in a household with married parents, that evidence is limited to externalizing behavior problems, and is explained by the financial and social capital families have access to. These findings support previous research that uncovered potential associations between children’s behavior and their closeness to their biological fathers [36,58]. For example, in our results, parental closeness was a significant factor in explaining away differences between youth in stepmother families and those with married biological parents. Marriage, whether because of mechanisms inherent to the structure itself or because of the privileges marriage enjoys as a celebrated historical-cultural object, has a notable protective association with children’s behavior, at least among biological parents. Youth in cohabiting biological families had greater internalizing behavior problems before accounting for financial and social resources, and worse externalizing behavior problems even after taking such resources into account. The findings for internalizing behavior problems provide some evidence supporting literature that claims associations between family structure and child outcomes are best explained by mechanisms connected to family structure, such as access to physical or social resources, than by the structure itself [15,16,17,18]. However, our findings concerning externalizing behavior problems still suggest components of family structure may be related directly to child outcomes. The fact that neither type of behavior problem differed for youth in stepmother and cohabiting biological families suggests that marriage is not a universal panacea, although we note it is difficult with these data to parse out the effects of marriage in stepfamilies. Still, it is notable that marriage seems to be most powerful when considering children’s biological parents. Perhaps this could be explained by cohabitating families being considered an unstable family environment, making them similar to stepmother families. Children from married biological families may occupy family environments that undergo less stress regarding the status and stability of their family. Married families also tend to have a higher income than other family structures, which is also related to family stress [3,5]. With higher incomes and greater social closure, families have more access to both physical and social resources, and in turn their children’s behavior problems decrease. 

An interesting finding in our study is the results of drug use, for both internalizing and externalizing behavior problems. We found higher use of drugs was related to fewer behavior problems. It is possible that our cross-sectional data was unable to identify a pattern where youth in more stressed family environments are more likely to turn to drugs to cope.

### 4.1. Limitations

In addition to the cross-sectional nature, our analyses have a few other limitations. To create a large sample of stepmother families, we chose to combine both cohabiting and married stepmothers. While this helped produce more accurate and trustworthy results, it does affect our ability to extend questions about marriage to stepmother families. In addition, it is possible that some negative associations with living in a stepmother family fail to reach conventional levels of statistical significance because of the small sample size. This seems most likely for the bivariate model comparing stepmother families and married biological families on internalizing behavior problems; in other models and other comparisons, the coefficients are small enough that we suspect lack of statistically significant findings is legitimate rather than a statistical artifact

### 4.2. Messages for Policymakers and Parents

It is important for parents to understand the influence the family environment has on children’s behavior. Cohabiters are a particularly interesting group to study in the UK, given that many more parents choose this family type there than in other contexts such as the US [52]. Because cohabitation is relatively normative in the UK, it is possible cohabiting parents do not see compelling reasons to marry; however, our findings suggest, along with a long history of research on family structure [1,2], that marriage between biological parents comes with associated benefits for children, at least in terms of their behavior. This may be a persuasive argument for some parents. If, on the other hand, cohabiters are delaying marriage rather than avoiding it, perhaps suggested by the relatively low socioeconomic status of the cohabiters we study here, our findings might be directed toward policymakers who could reduce financial barriers to marriage that would allow more couples to make that choice. Similarly, if these differences between married and cohabiting families are driven exclusively by access to financial and social resources, rather than anything inherent to the family structures themselves, similar interventions removing such barriers would also potentially benefit children’s development.

By contrast, we found little evidence for the idea that role conflict or lack of clarity around roles for stepmother families was resulting in family environments or processes that negatively influence children’s behavior in those families, especially when those families enjoyed similar structural positions as married biological parents did in terms of access to physical and social resources. While aid in helping stepmothers navigate their new roles, including training or classes, might be beneficial in general, our findings suggest the best ways for policymakers to invest in stepmother families are similar to the ways they might support cohabiting families: providing additional support that would increase or open up access to financial and social resources. Results from our models join the body of research underscoring the importance of policymakers considering family environment and its influence on children’s behavior and other key outcomes. 

## 5. Conclusions

While our findings provide intriguing information on the competing and complimentary roles of marriage and access to biological parents in children’s lives, there is still much we do not know. For example, this study focuses exclusively on children in the UK, which has a higher proportion of cohabiting biological parents than similar countries like the US or Australia. Future research should look into other countries that are earlier in the second demographic transition and where cohabitation might still be viewed as strongly nonnormative, such as in East Asia [59]. We also focused on behavior in early adolescence here; it is important to know if our findings concerning marriage and access to biological mothers apply to the “tender years” or to later adolescence when youth are starting to individuate from families. Large datasets like the MCS and similar also provide opportunities to build off the foundational findings we provide here and extend inquiries to longitudinal analyses that could follow children as they enter stepmother families or as their cohabiting parents marry to see if we are actually observing family environment differences or selectivity effects. While we focus on married parents here, fully understanding how stepmothers conceptualize and fill their parenting roles, and how those actions influence child outcomes, will require more research, including but not limited to comparing stepmothers to single mothers, which would provide the opportunity of comparing gendered explanations concerning roles to explanations that focus on access to resources. We also welcome samples that provide larger numbers of stepmother families that would allow for finer tests of the marriage hypothesis and the ability to test moderating effects. Regardless, our results are a useful jumping off point for further exploring the complex family environments that shape children’s lives.

## Figures and Tables

**Table 1 ijerph-19-16543-t001:** Measurement and Descriptive Statistics.

	Description	Mean/Proportion	Standard Error for	Range
Dependent variable				
Internalizing Behaviors	Childs internalizing behavior score	3.681	0.033	0–19
Externalizing Behaviors	Childs externalizing behavior score	4.239	0.034	0–20
Independent variables Family Structure				
Stepmother Families	Children who live with one biological parent and a stepmother	1%	0.001	
Biological Cohabitating Parents	Children who live with biological parents who are cohabitating	8%	0.003	
Biological Married Parents	Children who live with biological parents who are married	91%	0.003	
Family Closeness				
Father Closeness	Child’s reported closeness to biological father	2.744	0.011	0–4
Mother Closeness	Child’s reported closeness to biological mother	3.211	0.009	0–4
Parental Reported Closeness	Closeness to child as reported by the parent	3.342	0.007	0–4
Income				
Owns Home	Does the family owns their home 0 = No 1 = Yes	31% 69%		0–1
Income	Family’s income	3.217	0.014	1–5
Receives Child Benefits	Family has received income from the government 0 = No 1 = Yes	98% 2%		0–1
Parental Health:				
Parental Smoking	Does parent smoke 0 = No 1 = Yes	81% 19%		0–1
Parental Drinking	Does the parent drink Drinks alcohol Never drinks alcohol	23% 77%		0–1
Parental Health	Parental reported health	2.414	0.011	1–5
Parental Depression	Does the parent have severe depression	4.302	0.042	0–24
Child Health				
Sibling Number	Number of sibling Cohort Member has	1.538	0.011	0–7
Child Health	Child’s self-reported health	2.523	0.009	1–5
Children’s Extracurriculars	Is child involved in extracurriculars (clubs/sports) 0 = No 1 = Yes	3% 97%		0–1
Childs Drug Use	How often does child use drugs	0.772	0.010	0–4
Family Stress				
Family Stress	Family stress scale	0.439	0.006	0–4

Note. Data source is Millennium Cohort Study Sweep 6, n = 7, 182.

**Table 2 ijerph-19-16543-t002:** Means/Proportions by Family Structure.

Variable	Stepmother	Biological Married	Biological Cohabitating
n	77	6521	589
Internalizing	3.913	3.297	3.939
Externalizing	5.132	3.764	4.724
Parental Closeness			
Father Closeness	3.156	3.112	2.937
Mother Closeness	1.974	3.289	3.188
Parental Reported Closeness	3.172	3.349	3.283
Financial Resources			
Owns Home			
No	34%	15%	40%
Yes	66%	85%	60%
Income	2.766	3.676	3.327
Receives Child Benefits			
No	94%	98%	97%
Yes	6%	2%	3%
Parental Health			
Parental Smoking			
No	66%	89%	68%
Yes	34%	11%	31%
Parental Drinking			
Drinks alcohol	16%	23%	18%
Never drinks alcohol	84%	77%	82%
Parental Health	2.545	2.288	2.449
Parental Depression	4.222	3.744	4.656
Child Variables			
Sibling Number	1.987	1.599	1.392
Child’s Health	2.468	2.456	2.54
Children’s Extracurriculars			
No	1%	2%	3%
Yes	99%	98%	97%
Child’s Drug Use	1.038	0.632	0.799
Additional Variables			
Family Stress	0.429	0.386	0.403

Note. Data source is Millennium Cohort Study Sweep 6, n = 7, 182.

**Table 3 ijerph-19-16543-t003:** Ordinary Least Squares Regression of Internalizing Behavior on Family Structure (Model 1), Parental Closeness, (Model 2) Financial Resources (Model 3), Parental Health (Model 4), Child Health (Model 5), Family Stress (Model 6) and Full Model (Model 7). Stepmother families are the reference group.

Variable	Model 1	Model 2	Model 3	Model 4	Model 5	Model 6	Model 7
Biological Cohabitating Families	0.026 (0.384)	−0.284 (0.390)	0.268 (0.377)	0.008 (0.374)	−0.0004 (0.376)	0.045 (0.381)	−0.275 (0.375)
Biological Married Families	−0.615 (0.363)	−0.840 (0.370)	−0.146 (0.357)	−0.385 (0.355)	−0.624 (0.355)	−0.584 (0.360)	−0.463 (0.357)
Father Closeness		−0.488 *** (0.057)					−0.300 *** (0.057)
Mother Closeness		−0.201 *** (0.065)					0.135 * (0.063)
Parental Closeness		−0.360 *** (0.057)					−0.288 *** (0.054)
Owns Home			−0.377 *** (0.116)				−0.226 (0.119)
Income			−0.434 *** (0.033)				−0.255 *** (0.043)
Receives Child Benefits			0.059 (0.250)				−0.038 (0.260)
Parent Smoking				0.428 *** (0.111)			0.225 *** (0.112)
Parent Drinking				0.453 *** (0.092)			0.116 (0.100)
Parent Health				0.282 *** (0.040)			0.150 *** (0.041)
Parent Depression				0.212 *** (0.011)			0.191 *** (0.011)
Sibling Number					0.169 *** (0.035)		−0.148 *** (0.043)
Child Health					0.638 *** (0.042)		0.433 *** (0.043)
Children’s Extracurriculars					−10.924 *** (0.267)		−10.755 *** (0.262)
Child Drug Use					−0.166 *** (0.041)		−0.249 *** (0.041)
Family Stress Scale						0.739 *** (0.062)	0.186 ** (0.067)
Constant							

* *p* < 0.05, ** *p* < 0.01, *** *p* < 0.001. Note. Data source is Millennium Cohort Study Sweep 6, n = 7, 182.

**Table 4 ijerph-19-16543-t004:** Ordinary Least Squares Regression of Externalizing Behavior Problems on Family Structure (Model 1), Parental Closeness, (Model 2) Financial Resources (Model 3), Parental Health (Model 4), Child Health (Model 5), Family Stress (Model 6) and Full Model (Model 7). Stepmother families are the reference group.

Variable	Model 1	Model 2	Model 3	Model 4	Model 5	Model 6	Model 7
Biological Cohabitating Families	−0.409 (0.386)	−0.073 (0.386)	−0.188 (0.377)	−0.552 (0.381)	−0.130 (0.378)	−0.394 (0.384)	0.051 (0.375)
Biological Married Families	−10.369 *** (0.364)	−0.921 * (0.365)	−0.873 * (0.357)	−10.216 *** (0.362)	−10.014 ** (0.357)	−10.346 *** (0.363)	−0.320 (0.357)
Father Closeness		−0.177 *** (0.057)					0.008 (0.057)
Mother Closeness		−0.234 *** (0.064)					−0.240 *** (0.063)
Parental Closeness		−0.859 *** (0.056)					−0.782 *** (0.054)
Owns Home			−0.510 *** (0.116)				−0.242 * (0.120)
Income			−0.452 *** (0.033)				−0.381 *** (0.043)
Receives Child Benefits			−0.175 (0.253)				−0.199 (0.258)
Parent Smoking				0.860 *** (0.113)			0.417 *** (0.113)
Parent Drinking				0.226 * (0.094)			0.011 (0.100)
Parent Health				0.216 *** (0.040)			0.122 ** (0.041)
Parent Depression				0.177 *** (0.011)			0.448 *** (0.011)
Sibling Number					0.308 *** (0.035)		−0.086 * (0.042)
Child Health					0.255 *** (0.042)		0.041 (0.043)
Children’s Extracurriculars					−0.686 ** (0.268)		−0.326 (0.261)
Child Drug Use					0.584 *** (0.041)		0.491 *** (0.041)
Family Stress Scale						0.544 *** (0.063)	0.027 (0.066)
Constant							

* *p* < 0.05, ** *p* < 0.01, *** *p* < 0.001. Note. Data source is Millennium Cohort Study Sweep 6, n = 7, 182.

**Table 5 ijerph-19-16543-t005:** Ordinary Least Squares Regression of Internalizing Behavior Problems on Family Structure (Model 1), Parental Closeness, (Model 2) Financial Resources (Model 3), Parental Health (Model 4), Child Health (Model 5), Family Stress (Model 6) and Full Model (Model 7). Cohabiting families with both biological parents are the reference group.

Variable	Model 1	Model 2	Model 3	Model 4	Model 5	Model 6	Model 7
Stepmother Families	−0.026 (0.384)	0.284 (0.390)	−0.268 (0.377)	0.008 (0.374)	0.000 (0.376)	−0.045 (0.381)	0.275 (0.375)
Biological Married Families	−0.641 *** (0.137)	−0.556 *** (0.137)	−0.414 ** (0.138)	−0.392 ** (0.133)	−0.624 *** (0.135)	−0.629 *** (0.136)	−0.187 (0.133)
Father Closeness		−0.488 *** (0.057)					−0.300 *** (0.057)
Mother Closeness		0.201 ** (0.065)					0.135 * (0.063)
Parental Closeness		−0.361 *** (0.057)					−0.288 *** (0.054)
Owns Home			−0.377 *** (0.166)				−0.226 (0.119)
Income			−0.434 *** (0.033)				−0.255 *** (0.043)
Receives Child Benefits			0.059 (0.250)				−0.038 (0.260)
Parent Smoking				0.428 *** (0.111)			0.225 * (0.112)
Parent Drinking				−0.453 *** (0.092)			−0.116 (0.100)
Parent Health				0.282 *** (0.040)			0.150 *** (0.041)
Parent Depression				0.212 *** (0.011)			0.191 *** (0.011)
Sibling Number					0.169 *** (0.035)		−0.148 *** (0.043)
Child Health					0.638 *** (0.043)		0.433 *** (0.043)
Children’s Extracurriculars					−10.924 *** (0.267)		−10.755 *** (0.262)
Childs Drug Use					−0.166 *** (0.041)		−0.249 *** (0.041)
Family Stress Scale						0.739 *** (0.063)	0.186 ** (0.067)
Constant							

* *p* < 0.05, ** *p* < 0.01, *** *p* < 0.001. Note. Data source is Millennium Cohort Study Sweep 6, n = 7, 182.

**Table 6 ijerph-19-16543-t006:** Ordinary Least Squares Regression of Externalizing Behavior Problems on Family Structure (Model 1), Parental Closeness (Model 2), Financial Resources (Model 3), Parental Health (Model 4), Child Health (Model 5), Family Stress (Model 6) and Full Model (Model 7). Cohabiting families with both biological parents are the reference group.

Variable	Model 1	Model 2	Model 3	Model 4	Model 5	Model 6	Model 7
Stepmother Families	0.409 (0.386)	0.073 (0.386)	0.188 (0.377)	0.552 (0.381)	0.130 (0.378)	−0.94 (0.384)	−0.051 (0.275)
Biological Married Families	−0.960 *** (0.137)	−0.849 *** (0.134)	−0.685 *** (0.138)	−0.664 *** (135)	−0.884 *** (0.135)	−0.9512 *** (0.137)	−0.371 ** (0.133)
Father Closeness		−0.177 ** (0.057)					0.008 (0.057)
Mother Closeness		−0.234 *** (0.064)					−0.240 *** (0.063)
Parental Closeness		−0.859 *** (0.056)					−0.782 *** (0.053)
Owns Home			−0.510 *** (0.116)				−0.242 * (0.119)
Income			−0.452 *** (0.033)				−0.381 *** (0.043)
Receives Child Benefits			−0.175 (0.253)				−0.199 (0.258)
Parent Smoking				0.860 *** (0.113)			0.127 *** (0.113)
Parent Drinking				−0.226 * (0.094)			0.011 (0.100)
Parent Health				0.216 *** (0.094)			0.122 ** (0.041)
Parent Depression				0.177 *** (0.011)			0.148 *** (0.011)
Sibling Number					0.308 *** (0.035)		−0.086 * (0.042)
Child Health					0.255 *** (0.042)		0.041 (0.043)
Children’s Extracurriculars					−0.686 ** (0.268)		−0.326 (0.261)
Childs Drug Use					0.584 *** (0.041)		0.491 *** (0.041)
Family Stress Scale						0.544 *** (0.063)	0.027 (0.066)
Constant							

* *p* < 0.05, ** *p* < 0.01, *** *p* < 0.001. Note. Data source is Millennium Cohort Study Sweep 6, n = 7, 182.

## Data Availability

MCS data may be acquired through contract with the UK Data Service (https://beta.ukdataservice.ac.uk/datacatalogue/series/series?id=2000031, accessed on 1 January 2022).

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
