# Peer review of "Comparing Children’s Behavior Problems in Biological Married, Biological Cohabitating, and Stepmother Families in the UK"

_ijerph, 2022, doi:10.3390/ijerph192416543_

Round 1

Reviewer 1 Report

Thank you for the opportunity to review this insightful article. There are a few suggestions to enhance the manuscript:

1. Ethics - There is no mention except at the end of the article as to who granted ethical approval for this study. 

2, Measures - It is not clear if the measures used were constructed prior to the data collection or were designed by the authors of this study? Were there any reliability, validity studies conducted upon them. Perhaps the authors could elaborate upon this point. They should also be considered in the limitations section. 

3. Discussion - The discussion could be enhanced more by discussing the results/findings more with the literature. 

4. Page 14 line 395 refers to Models 5 and 6 however, I think this should state Models 4 and 5 as Model 6 is referred to in the next paragraph line 406.

Reviewer 2 Report

Journal: Int. J. Environ. Res. Public Health 2022, 19

Title: Comparing Children’s Behavior Problems in Biological Married, Biological Cohabitating, and Stepmother Families in the UK

Author/s: M. Rachél Hveem, Samuel C.M. Faulconer and Mikaela J. Dufur

Summary: The study adopted the Millennium Cohort Study (MCS) sweep 6 to compare children’s behavior problems in biological married, biological cohabitating, and stepmother families. The results indicate that the family structure has no significant relationship with regard to children’s internalizing behavior problems. On the other hand, the results indicate that children living with stepmother and biological cohabiting parents have more externalizing behavior problems than those who live with biological married parents. The MCS is a longitudinal study that looked at children born in UK in 2000. Sweep 6 indicates the 18,818 children who were 14 at the time data was collected. After applying the exclusion criteria, the final sample is of 7,182, with 6,521 two-parent married families, 589 two-parents cohabiting families, and 77 two-parent stepmother families.

Questions:

In section “1 Introduction”, line 26-37, there is the need to clarify the relationship (either in quantitative or qualitative ways) between the construal/concept of “family environment” and “family structure”.

In section “1 Introduction”, line 43, it is not clear the meaning of the expression “more normative”. Please, clarify.

In section “1 Introduction”, line 45-48, it is not clear if you (or the research) had a (pre-intended) aim of highlighting the importance of living with a biological mother. Please, clarify and, more importantly, please state what you mean with the expression “importance” (line 47).

Generally speaking, the section “1 Introduction” uses 8 references to support its arguments. Six out of eight are from 1987, 2004, 2005, 2005, 2006, 2009. They are quite dated.

Moreover, it refers only to supportive literature. Therefore, the is the need to understand if there are no available research that states the contrary on authors’ arguments.

In section “1.1 Children’s Behavior Problems”, line 54-55, there is the need to strongly support the argument “Family structure has a significant effect on children’s behavior problems”.

In section “1.2 Biological Family Environment”, line 60-62, there is the need for references supporting the statement.

In section “1.5 Stepmother Family Environment”, statements from line 128 to 136 need proper supportive references.

In section “1.6 Current Study”, line 174-177, there is the need to rephrase the sentence since there might be a mistake (there is the repetition of the expression, “In the United States…”).

In section “1.6 Current Study”, line 183-186, it is not clear on what basis you “predict” and, most importantly, why. Finally, in the mentioned prediction, it is written “…because they have access to biological mothers”. Therefore, this statement implies a causality (why? Because…) and, most importantly, the identification of a key explaining factor, that is the “access” to biological mothers. What does “access” mean, firstly? Secondly, what does the overall conclusion that children from cohabitating biological families report in between children’s behavior problem because they have access to biological mother means? There is a confusion between a descriptive and explanatory (or at least hypothetical) dimension.

In section “2.2 Measures”, line 212-214, there is the need for a clarification in the used language. It states that internalizing behavior problem scale (first <variable>) is made up of 10 <variables> and that the externalizing behavior problem is made up of 12 <variables> (second <variable>). If IBP and EBP are variables, then necessarily they can not be made up of more variables. On the other hand, if they are actually made up of variables, then IBP and EBP are concepts and/or construal.

In section “2.2 Measures”, line 215-221, there is the need for a clarification in the used language (see above).

In section “2.2 Measures”, line 222-229, there is the need for a clarification in the used language (see above).

In section “2.2. Measures”, line 237-243, first theoretical block, one could wonder if “closeness” is a self-evident concept that needs no description/explanation. Moreover, how can you assure that this measurement actually measures “closeness”?

In section “2.2 Measures”, line 261-268, there is the need to clarify how you “created” the “family stress scale”. Why did you rely upon that five <variables> and no other variables?

In section “4.1 Results Summary”, line 473-474, I think that the question you posed should be reformulated or, at least, better clarified in other sections.

In section “4.1 Results Summary”, line 477-479, the following statements “that children’s behavior can be predicted based on their closeness to their biological father” is pretty strong, and it does not seem to be supported by your data.

In section “4.1 Results Summary”, line 479, please clarify the following expression: “Our study found this to be true”.

In section “4.1 Results Summary”, line 481-483, I wonder if the correct verb in the statement (“to have”) is correct. As explained in further details in the comment section, the verb “to have” might suggest a reification which takes for granted marriage as a “natural thing” rather than a “historical-cultural object”.

In section “4.3 Messages for Policymakers and Parents”, line 519-520, consider the following statement: “marriage between biological parents has protective effects for children, at least in terms of their behavior”. I have a question: Does the marriage, itself and in general and defined as two parents being married, express this protective effects?

Comment and Evaluation:

I thank the Editor for having me review this research and I thank the Author/s for their work.

Overall, I found the research interesting.

However, I raised several (mainly) epistemological concerns, that it is about how concepts/construal are defined/operationalized and how statements are justified and therefore written.

I suggest the paper to be reviewed.

Reviewer 3 Report

The work presents an interesting topic that, in general terms, is adequate and meets the requirements of a work of these characteristics. Nevertheless, some issues to be considered by the authors are discussed below.

Although it is understood by context that the authors refer to heterosexual couples, both when speaking of biological families and stepfamilies, this should be indicated in some way.

It is not clear in the method whether the authors have participated in any way in the study from which they obtain the data (the Millennium Cohort Study) or whether the data have been obtained passively. Nothing is specified about the study or when it was carried out. In fact, in the table footnote in Tables 2 to 6 it is indicated that the source of the data is the National Longitudinal Study of Youth, 1997 while the data in Table 1 are obtained from the Millennium Cohort Study. Is it the same study?

Reliability and consistency data for the instruments should be indicated.

In Table 2, the figures for participants by family structure should be indicated by (n) instead of (N).

Reviewer 4 Report

The manuscript presents an interesting study on children’s behavior problems in different types of families, but it has some issues that need to be resolved before the publication.

Introduction:

- The whole section favors environment where a child lives with married biological parents. The authors use very one-sided references – studies that found these kinds of influences. Also, we can not find any references about “non-healthy” marriages and their influence on children. Please, revise according to this.

- Lines 38-48: this part would be more suitable for the end of Introduction (“Current study”).

- Subsections 1.1. – 1.5.: I am not sure if division into subchapters is needed.

- Line 84: this statement is true for some of the studies, not for all the body of research. Please add a reference here. In the next lines of the same subsection, please add some studies that found different results – not every study reports such strong influence of marriage.

- Lines 128-132: this is a very strong generalization! Please add a reference and write the statements less deterministically.

Materials & methods:

- Please add the authors of the used instruments (SDQ).

- Table 1: Please move it to the beginning of the Results. The table could also be prepared in a more concise way.

Results:

- Please prepare the tables in a more concise way (they are too large, with very big gaps).

Discussion:

- Subsection 4.1.: I do not think it is necessary to divide it from the main chapter.

- Lines 517-520: this is a very deterministic statement.

Conclusions:

- It must be prepared as a separate section.

Round 2

Reviewer 2 Report

The article can be accepted

Reviewer 4 Report

Introduction:

-       It would seem more sense for me to make the introduction more openly. So, in the sense that you are interested in how the family structure is related to children's problems. You can then present the findings of studies that confirm the positive effects of married biological parents, as well as those who find that there are no differences in children's outcomes according to the family structure. I would only like to emphasize that I do not expect to present literature that says that children who do not live with married biological parents have less problems, but one that finds that there are no differences in children's outcomes according to the family structure. For example, I suggest the authors to read the meta-study by Biblarz & Stacey (2010). know it's not exactly the same topic but you can see the direction of thinking about the family structure and its effects on children.

Results:

-       The tables are still very large. I do not see the need to include so many empty lines and so large gaps.

Discussion:

-       Please try to provide some additional explanations according to the possible new references in the Introduction.
